# The Effect of Fermented Grains (koji) on Physicochemical and Sensory Characteristics of Chicken Breasts

**DOI:** 10.3390/foods12183463

**Published:** 2023-09-17

**Authors:** Jaehyun Jeong, Seungtak Jeon, Jiseon Lee, Mi-Yeon Lee, Kwang-Hee Lee, Chi-Kwang Song, Mi-Jung Choi

**Affiliations:** 1Department of Food Science and Biotechnology of Animal Resources, Konkuk University, Seoul 05029, Republic of Korea; wjdwogus14031@naver.com (J.J.); tmdxkr322@gmail.com (S.J.); ezeess91@gmail.com (J.L.); myli00@nate.com (M.-Y.L.); 2Sangkyung F&B, 1100, Dongan-ro, Yeonmu-eup, Chungchenongnam-do, Nonsan-si 33011, Republic of Korea; rokmc6310@skfnb.net (K.-H.L.); skf0613@skfnb.net (C.-K.S.)

**Keywords:** chicken breast, koji, physicochemical property, sensory analysis

## Abstract

This study investigated the tenderizing and flavor-enhancing effects of koji, a fermented grain cultured with a single microorganism, on chicken breasts during curing. Chicken breasts were cured with different ingredients, including 4% (*w*/*w*) curing agent (GC), 5% (*w*/*w*) *Aspergillus oryzae* with rice (FR), *A. oryzae* with soybean (FS), and *Bacillus subtilis* with soybean (BS) for 4 h at 4 °C prior to cooking. After the superheated steam procedure, all samples were cooked in a convection oven, and their physicochemical properties were analyzed. Koji-treated samples exhibited significantly higher expressible moisture due to the degradation of the protein matrix (*p* < 0.05). Texture profile analysis showed that the tenderness of koji-treated samples was significantly higher than that of GC (*p* < 0.05). Furthermore, koji-treated samples were regarded as tenderer, and they were preferred over GC (*p* < 0.05) in the sensory evaluation. Principal attributes analysis revealed that the overall preference for koji-treated samples was highly correlated with umami, juiciness, and tenderness (*p* < 0.05). Overall, this study provides insights into applying koji as a potential curing treatment to improve the eating quality of chicken breasts. Koji can be used as a novel technology in the food industry to improve taste and tenderness simultaneously.

## 1. Introduction

The consumption of chicken meat has significantly increased due to its outstanding nutrient composition, including high protein, low fat content, low cholesterol, and comparatively low cost as a protein source compared with red meat [1,2]. With the increasing awareness of health and nutritional value, its recognizable nutritional benefits can make customers feel that chicken meat helps them maintain or enhance their healthy diet [3,4]. However, chicken breasts have a dry and tough texture due to their low-fat content. Although the primary considerations for meat products are the amount of fat and color of meat in the purchase decision phase, the most important factors after cooking are flavor and tenderness [5,6,7,8].

Curing is a common method to improve tenderness and the preservation period by soaking meat in a salt solution [9]. Phosphates and sodium chloride, the most prevalently used, improve the protein solubility, thereby increasing water-holding capacity and shortening the sarcomere length related to meat tenderization [9,10,11]. Further tenderization can be performed using enzymatic treatment [8]. As meat tenderizers, proteolytic enzymes are best suited for degrading collagen and elastin in connective tissue at relatively low temperatures [12]. Plant proteases, such as papain (from the latex of papaya plant, *Carica papaya*), bromelain (from pineapple, *Ananas comosus*), and ficin (from the latex of fig, *Ficus glabrata*, *Ficus anthelmintica*, and *Ficus laurifolia*), and microbial proteases from *Aspergillus oryzae* and *Bacillus subtilis* have been adapted to exogenous proteases for meat tenderization [8]. The United States Department of Agriculture has approved that they can be generally regarded as safe [13,14]. Plant proteases have high activity, but are low in specificity, resulting in problems of over-tenderization [8]. The addition of inhibitors is one of the most reasonable ways to compensate for this, but some inhibitors such as ascorbic acid have the disadvantage of causing odor and deteriorating the quality of the product [15,16]. Furthermore, it can be an inconvenient process compared to microbial proteases. The microbial proteases from *A. oryzae* and *B. subtilis* can be applied as meat tenderizers in households and the food industry without the risk of over-tenderization, as their activity is self-limiting [17,18].

Koji, a fermented grain using a single microorganism, is used as a starter for food fermentation due to its enzymatic activity in alcoholic beverages (makgeolli and sake) and fermented soybean products (miso, doenjang, and soy sauce) [19,20]. Compared with nuruk, fermented grains containing a wide range of airborne microorganisms, koji has stronger enzyme activity due to the purity of the organic network [21]. Enzyme activity affects not only the flavor but also the nutritional components, texture, and color of the fermented products [19,21,22]. Furthermore, koji directly influences the taste and flavor of products by producing sugar and amino acids [23]. It can be hypothesized that adding koji to meat products during curing could affect tenderization and enhance flavor.

Few studies have examined the effects of koji to improve the eating quality of meat products. Hiemori-Kondo et al. [24] investigated the physiological and sensory properties of sika deer marinated in fruits, ginger, shio-koji, amazake, and miso (*A. oryzae*). Among them, meat treated with miso and shio-koji exhibited low cooking loss and rupture stress, and in sensory evaluation, meats treated with shio-koji, miso, and ginger were tender and preferred. Furthermore, Kim et al. [25] demonstrated the capacity of koji as soy sauce to improve the tenderness of chicken breast. However, there is very little information available about the effect of koji between rice and soybean, especially on the physicochemical and sensory properties of chicken breast. Therefore, this study aimed to investigate the effects of different types of koji on the physicochemical properties of chicken breast meat. We focused on tenderness for all properties to meet market needs.

## 2. Materials and Methods

### 2.1. Materials

Fresh chicken breasts (130–180 g; Harim Industrial Co., Ltd., Seoul, Republic of Korea) were purchased from a local market. Goldchickide (GC) is composed of refined salt, white pepper powder, ginger flour, and garlic flour as a commercial curing agent, and fermented grains were purchased from Sangkyung F&B (Nonsan, Republic of Korea). Rice with *A. oryzae* (FR), soybeans with *A. oryzae* (FS), and soybeans with *B. subtilis* (BS) were selected as fermented grains for the curing solution.

### 2.2. Preparation of Cured Chicken Breasts

A mixture of curing solutions was created as chicken breast:water:GC (1000:100:4, by weight), and FR, FS, and BS were supplemented with 5% (*w*/*w*) curing agent. Raw chicken breast samples were used as controls (C). Chicken breasts packed with curing solutions were tumbled using a curing machine (STX-1000-CE; Mercantile Station 2, Lincoln, NE, USA). The tumbling conditions were 0 °C for 60 min under vacuum pressure. The samples were stored at refrigerated temperature (4 °C) for 4 h prior to cooking.

### 2.3. Conditions of Superheated Steam (SHS) and Convection Oven Cooking

SHS treatment was conducted in a superheated steam oven (DC Quto QF-5200C; Naomoto, Japan) with a preheated oven temperature of 100 °C and steam temperature of 250 °C for 1 min. After the SHS process, the chicken breasts were rapidly cooled on an ice pack for 3 min to prevent moisture loss, stored in a refrigerator, and packed in plastic bags before cooking in a convection oven.

Convection oven cooking was conducted using a combined oven (EON-C326; SK Magic, Seoul, Republic of Korea). The samples were cooked until the core temperature reached at least 75 °C in the oven preheated to 200 °C. Temperature was measured using a probe thermometer (PT-305; Foronemillion, Seoul, Republic of Korea). After cooking, the samples were cooled at room temperature (24 °C) for 10 min and stored in a refrigerator at 4 °C till analysis.

### 2.4. Visible Appearance and Color Profile Analysis

The external and internal appearances of chicken breasts were observed using a digital camera (a350; Sony, Tokyo, Japan). The color values were measured on the medial surface of each sample using a color reader (CR 400, Konica Minolta Sensing Inc., Tokyo, Japan) calibrated with a white standard plate (CIE L*, lightness, 97.83; CIE a*, redness, 0.43; CIE b*, yellowness, 1.98). The instrument was directly placed at different points on the chicken breast surface.

### 2.5. Water Content

The water content of chicken breasts was determined using the heat-drying method [26]. The cooked chicken breasts were weighed and dried at 105 °C using a drying oven (OF-105; Daihan Scientific Co., Ltd., Gangwon-do, Republic of Korea) until reaching a constant weight. Water content was calculated as follows:Water content (%) = [(W_1_ − W_2_)/ W_1_] × 100 (1)
where W_1_ and W_2_ are the initial and final weights of the sample, respectively.

### 2.6. Cooking Loss

The weight of chicken breasts before and after cooking was measured, and the cooking loss was calculated using the following formula:Cooking loss (%) = [(W_1_ − W_2_)/ W_1_] × 100 (2)
where W_1_ and W_2_ are the weights before and after cooking, respectively.

### 2.7. pH

To measure the pH, chicken breasts (2 g) were homogenized with distilled water (18 mL) for 1 min. The pH was measured using a pH meter (S-220; Mettler Toledo Co., Zurich, Switzerland). Readings were obtained in triplicate for each sample.

### 2.8. Uptake of Curing

The weight of chicken breasts before and after curing was measured, and the uptake of the marinade was calculated using the following formula:Uptake of curing (%) = [(W_2_ − W_1_)/ W_2_] × 100 (3)
where W_1_ and W_2_ are the weights before and after curing, respectively.

### 2.9. Expressible Moisture (EM)

The ability of chicken breast products to retain moisture was assessed according to Lee et al. [27], with some modifications. Approximately 1 g of minced chicken breast was placed into a 15 mL conical tube and centrifuged at 3000× *g* rpm for 10 min at 4 °C. EM was calculated using the following formula:EM (%) = [(W_2_ − W_1_)/W_2_] × 100 (4)
where W_1_ is the initial sample weight (g) and W_2_ is the weight of the sample after centrifugation (g).

### 2.10. Texture Profile Analysis (TPA)

Modifying the method of Cho et al. [28], the samples were divided in five. The second spots from the thicker part were selected for the texture analysis. Moreover, texture analysis of the samples was only conducted after cooling for 30 min at 25 °C [29].

TPA was performed using a texture analyzer (CT3; Brookfield Co., Middleboro, MA, USA). The samples (2 × 2 × 2 cm) were compressed twice to 50% of their original height at a speed of 2 mm/s and trigger load of 100 mN using a cylindrical probe (TA44, 4 mm diameter). The hardness, adhesiveness, cohesiveness, springiness, gumminess, and chewiness values were calculated from the force and time curves.

### 2.11. Sensory Test

Sensory testing was conducted to confirm the effect of the fermented grains on the appearance, scent, umami, juiciness, and tenderness of chicken breasts. Ten students were selected and trained in sensory testing. Two of these were designated as alternatives if panelists were unavailable for testing. For testing, the samples were stored at a refrigerated temperature (4 °C) for 24 h and heated in a microwave oven (MW25B; LG Electronics, Seoul, Republic of Korea) for 30 s. After that, the samples, including chicken skin, were provided at the same temperature (25 °C) and size of 2 × 2 × 2 cm (width × length × height) after a 3-digit random number was assigned to each. Warm water was provided to the panelists to rinse their mouths between tastings. The test was conducted using a 9-point scale (score test), for appearance, scent, umami, juiciness, tenderness, and overall preference, with the lowest scoring level expressed as 1 and the highest level as 9. This study was approved by the institutional review board (IRB) at Konkuk University (IRB approval number: 7001355-202302-HR-650).

### 2.12. Statistical Analysis

All reported values were derived from testing in triplicate (or more), and results are presented as mean ± standard deviation. Means were compared using one-way ANOVA followed by Duncan’s multiple range test (*p* < 0.05). Statistical analysis was performed using the SPSS software (version 24.0; SPCC Inc., Chicago, IL, USA).

## 3. Results and Discussion

### 3.1. Appearance and Color Measurement

Appearance and color, the first impressions of food, are crucial factors that affect consumer acceptance of poultry meat [30,31], and the results are shown in Figure 1 and Table 1. No intuitive differences were observed among the chicken breasts for external and internal appearance.

However, the color measurements of the chicken breasts differed significantly in terms of lightness (CIE L*) and redness (CIE a*). The color difference in chicken breasts, generally dominated by white muscle fibers, is due to the ratio to red muscle fibers [32]. In the CIE L* measurement, C (76.2 ± 3.0) and FS (74.8 ± 2.8) had the highest values, while in the CIE a* measurement, BS (5.1 ± 2.2) showed the highest value, which was significantly higher when compared to other chicken breasts (*p* < 0.05). In the CIE b* measurement, GC (20.0 ± 1.8), FR (19.6 ± 4.9), FS (19.9 ± 5.0), and BS (19.6 ± 2.9) were slightly higher than C (17.8 ± 3.3). Kim et al. [33] reported that an increase in the CIE b* value was due to the browning color of the curing treatments. Additionally, the synthesis of a brown polymer known as melanoidin and the greater concentration of acrylamide,, are primarily responsible for the CIE a* values of the products [34].

### 3.2. Water Content, Cooking Loss, and pH

The water content of chicken breasts supplemented with different curing treatments is presented in Table 2. Water content for all treatments ranged from 68.4 ± 0.6% to 70.1 ± 0.3% and thus did not significantly differ (*p* > 0.05).

In the cooking loss measurement (Table 2), GC (21.8 ± 1.8%) and FR (23.8 ± 0.9%) exhibited slightly higher values than those of C (24.0 ± 0.2%), FS (24.0 ± 0.0%) and BS (23.3 ± 0.9%) but there was no significant difference observed (*p* > 0.05). It is considered that the heating processes, i.e., SHS and convection oven, already led to sufficient loss during cooking. This is consistent with the results of Yusop et al. [35], who also demonstrated no significant effect on cooking loss of chicken breast when using different marinade treatments after dry heating and steam cooking for 15 min. Moreover, Augustyńska-Prejsnar et al. [36] indicated no effect on cooking loss when using acidic marinated chicken breasts. Lopez et al. [37] also demonstrated that over 1.0% (*v*/*v*) sodium chloride concentration in tenderizers had no significant effect on cooking loss and moisture retention of chicken breast meat.

pH is a necessary indicator that affects other physicochemical and quality parameters of meat products [38]. The pH values of cured chicken breasts are presented in Table 2. The pH values for all treatments ranged from 6.1 ± 0.1 to 6.2 ± 0.0. A significant decrease in pH values (*p* < 0.05) was observed in FR (6.1 ± 0.1), FS (6.1 ± 0.1), and BS (6.1 ± 0.0), all of which were treated with fermented grains, while C (6.2 ± 0.1) and GC (6.2 ± 0.0) did not exhibit significant differences. A pH value close to the isoelectronic point of myosin (5.4) leads to the lowest water-holding capacity due to the contraction of the sarcomere matrix [39]. However, the pH values of poultry, ranging from 6.1 to 6.2, have little impact on muscle tissue [40]. Thus, the difference in the pH values (<0.1 unit) might not be important in practice [41]. Furthermore, the decrease in pH was not concluded to be a contraction of protein tissue, but rather the destruction of the myofibrillar protein matrix itself by tenderization [42]. Similarly, koji-treated samples, which were closer to pH 5.4, exhibited lower hardness values in the present study.

### 3.3. Uptake of Curing

The curing values are depicted in Figure 2A. The treated samples (GC, FR, FS, and BS) exhibited a weight gain during the curing process, while C (−0.9 ± 0.4%) exhibited a weight loss. This is correlated with mass transfer, including salt diffusion from curing treatments to meat and water flow from meat to curing treatments [43]. Additionally, the curing agent diffuses into the tissue due to the exudation of soluble materials, such as proteins and taste components [44]. Significant weight gains (*p* < 0.05) were observed in FS (3.3 ± 0.4%) and BS (2.3 ± 0.5%). This indicates that FS and BS have higher penetration into chicken breasts. The ability of a curing agent to penetrate chicken breast is attributed to its absorption rate [9]. According to Xiong et al. [45], the increased absorption rate can also be linked to an increase in ionic strength, improving the interaction of the molecules with the protein matrix. Thus, koji-treated samples (FS and BS) effectively accelerated mass transfer as a curing treatment for chicken breasts, indirectly promoting tenderization.

### 3.4. Expressible Moisture

In this study, EM was used to trace the moisture content in chicken breasts. The EM of a protein system is determined by the release of free water under applied force [46]. Therefore, EM is related to the sensory evaluation of juiciness [47]. The EM of chicken breasts subjected to different curing treatments is presented in Figure 2B. The EM values ranged from 16.9 ± 2.8% to 24.3 ± 4.2%. A significant increase in EM values was observed in GC (20.0 ± 4.3%), FR (23.9 ± 1.1%), FS (24.3 ± 4.2%), and BS (21.4 ± 0.9%) compared to C (16.9 ± 2.8%). Specifically, koji-treated samples (FR, FS, and BS) had significantly higher EM values (*p* < 0.05). The tendency of EM to increase was due to the absorption during the curing process. Thus, the significantly higher values of FS and BS during curing indicated that more swelling occurred in chicken breasts. The swelling has been associated with higher electrostatic repulsion between myofilaments and depolymerization of myosin filaments, which results in space expansion under the myofilaments [48]. The increase in EM in enzymatically tenderized meat is due to changes in the myofibrillar protein matrix caused by exogenous proteolytic enzymes [49,50]. Furthermore, higher EM induces higher drip loss, whereas enzymes promote protein degradation and denaturation in muscles [51,52].

**Figure 2 foods-12-03463-f002:**
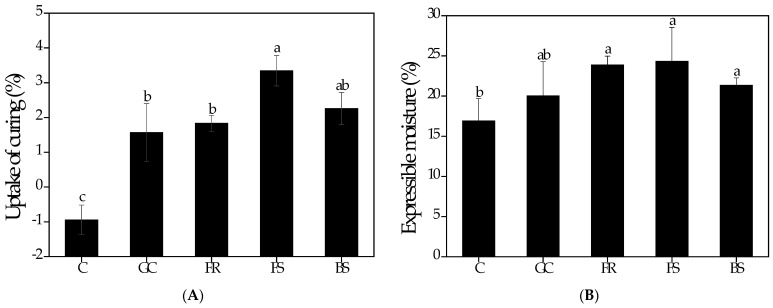
Uptake of curing (**A**) and expressible moisture (**B**) of chicken breasts with different curing treatments. C, treated by superheated steam and hot air oven; GC, treated with Goldchickide as curing agent by superheated steam and hot air oven; FR, treated with rice and fungi (*A. oryzae*) as curing agent by superheated steam and hot air oven; FS, treated with soybean and fungi (*A. oryzae*) as curing agent by superheated steam and hot air oven; BS, treated with soybean and bacteria (*B. subtilis*) as curing agent by superheated steam and hot air oven. ^a–c^ Means with different letters are significantly different based on one-way ANOVA followed by Duncan’s multiple range test (*p* < 0.05).

### 3.5. Texture Profile Analysis

TPA has been commonly applied in the characterization of food to measure textural parameters necessary for food acceptability [53]. The TPA values of chicken breasts subjected to different curing treatments are shown in Table 3. The result showed that the hardness of FR (56.8 ± 3.5 N), FS (52.8 ± 6.5 N), BS (57.7 ± 8.8 N), and GC (61.7 ± 9.6 N) was significantly lower (*p* < 0.05) than C (66.3 ± 4.1 N). Additionally, koji-treated samples (FR, FS, and BS) were significantly lower (*p* < 0.05) than those treated with GC. This is related to the lower pH, and higher EM values effectively decrease the hardness due to the denaturation of sarcoplasmic proteins [14]. Fencioglu et al. [54] reported that curing treatments containing curing agents and fermented grains reduced muscle fibril hardness, improving texture and taste. Furthermore, FS, a combination of NaCl and *A. oryzae*, had the lowest values (*p* < 0.05). Sullivan and Calkins [55] suggested that *A. oryzae* has the greatest fraction of salt-soluble proteins, implying enhanced protease activity in meat systems. Furthermore, proteolysis had a relatively positive effect on chicken breast tenderness. Based on a prior investigation, the protease activity of *A. oryzae* in soybean (311.2 units/g) was significantly higher (data not shown). Hong and Kim [56] also demonstrated that protease activity in soybean koji was significantly higher than that in rice koji because of the different degrees of protein content (*p* < 0.05).

The use of curing agents significantly affected the cohesiveness of chicken breasts. The cohesiveness of the treated samples, GC (0.4 ± 0.1), FR (0.4 ± 0.0), FS (0.4 ± 0.1), and BS (0.4 ± 0.0), was significantly higher (*p* < 0.05) compared to that of C (0.3 ± 0.0). Cohesiveness provides knowledge of viscoelastic attributes, such as tensile strength [57]. The increase in the cohesiveness of meat products is promoted by the release of the tenderizer due to the destructive effects on myofibrillar protein during curing [58].

Additionally, the treated samples, GC (8.9 ± 0.1), FR (8.5 ± 0.4), and BS (8.6 ± 0.1), exhibited significantly higher (*p* < 0.05) springiness values than C (8.00 ± 0.7). Meat springiness is attributed to the swelling of the fiber, which is apparent in the fiber diameter [57]. This is due to the denaturation of myosin and α-actinin, which affects this parameter [59].

### 3.6. Sensory Test

In contrast to physicochemical experiments, sensory evaluation of foods is critical because it analyzes the functioning of the senses and brain, focusing on people [60]. Thus, the results of the sensory analysis are crucial indicators of the quality of chicken breast and people who consume the food [61]. The sensory evaluation of chicken breasts subjected to different curing treatments is presented in Table 4 and Table 5. No significant differences were observed in the appearance and scent analysis for any treatments. The results showed that the curing agents had no intuitive adverse effects. The index of chicken breasts supplemented with koji, which correlated with taste and texture, showed significant differences in comparison to the control samples (*p* < 0.05). BS had the highest values for specific attributes, such as umami, juiciness, and tenderness (*p* < 0.05). Furthermore, koji-treated samples (FR and FS) were significantly higher than those treated with C and CG (*p* < 0.05).

In the intensity analysis of umami, BS (6.3 ± 0.9) scored the highest value, which was significantly higher than the others (*p* < 0.05). Furthermore, the values for FR (5.8 ± 1.3) and FS (5.4 ± 0.5) were significantly higher than those of C (3.6 ± 1.0) and GC (4.7 ± 1.1) (*p* < 0.05). Similarly, BS (6.5 ± 0.9) scored the highest value in the preference analysis of umami, which was significantly higher than the others (*p* < 0.05). FR (6.0 ± 1.3) and FS (5.5 ± 2.0) were also significantly higher than C (4.00 ± 0.94) and GC (5.1 ± 1.0) (*p* < 0.05). Fermented products have distinct flavor profiles due to the proteolytic activity of microorganisms that disintegrate proteins into peptides and free amino acids [62]. Furthermore, the fragments that are important flavor-active substances supply umami and desirable sensory qualities [63]. Chen et al. [64] also demonstrated that the taste of peptides derived from chicken protein via hydrolysis using *A. oryzae* and *B. subtilis* corresponded to umami using an electronic tongue and sensory evaluation.

Juiciness analysis also revealed that koji enhanced the sensory characteristics of chicken breasts. In the intensity measurement of juiciness, BS (6.6 ± 1.0) showed the highest value, which was significantly higher than the others (*p* < 0.05). Moreover, the koji-treated samples (FR; 5.7 ± 1.3) also scored significantly higher values than the others (*p* < 0.05). Meanwhile, FS (5.2 ± 1.6) exhibited a significantly lower value among the koji-treated samples (*p* < 0.05). This is associated with the excessive release of water from the protein matrix, which correlates with higher EM, as shown in the present study. The preference analysis of juiciness also indicated that BS (6.7 ± 1.5) scored the highest (*p* < 0.05). Juiciness is highly correlated with water retention [65,66]. This can be demonstrated using crosslinked polymer networks modulating the polymer–water affinity and crosslinking density [67]. The koji-treated samples, which showed significantly high uptake of curing and EM values, had more juiciness through sensory evaluation due to the migration of water and binding with the crosslinked polymer network. Chumngoen et al. [68] reported that increased EM in muscles improves juiciness and eventually increases the palatability of meat products.

Tenderness analysis of koji-treated samples revealed a significantly higher intensity and preference than C and GC (*p* < 0.05). This is associated with the degree of hardness in TPA, which is an instrumental analysis, demonstrating the variation in chicken breasts as evaluated by the sensory test results of the panelists [69]. It can be assumed that microbial proteases, including those produced by *A. oryzae* and *B. subtilis*, tend to preferentially break down myofibrillar proteins over collagen proteins while providing positive sensory outcomes [55]. Specifically, BS (6.3 ± 0.9) scored the highest value in intensity, which was significantly higher than the others (*p* < 0.05). Tenderness is necessary for determining the quality of meat products in sensory tests, and knowledge of myofibrillar protein fragmentation is essential for optimizing meat proteases and processing [70]. Due to the significantly higher values of BS in the sensory analysis of principal attributes, we can surmize that overall preference is also correlated with the umami, juiciness, tendernesssensory results, whereby BS (7.4 ± 1.1) exhibited the highest value, which was significantly higher than the others (*p* < 0.05).

## 4. Conclusions

This study demonstrates that koji is acceptable for enhancing the texture and flavor of chicken breasts. The koji-treated samples (FR, FS, and BS) had significantly higher EM due to the degradation of the protein matrix (*p* < 0.05). For TPA, the tenderness of koji-treated samples was significantly higher compared to GC (*p* < 0.05). Furthermore, no significant differences were observed in the cooking loss or water content. In the sensory test, the koji-treated samples were regarded as tenderer and preferred by the panelists over the GC (*p* < 0.05). Principal attributes analysis revealed that the overall preference for koji-treated samples was highly correlated with umami, juiciness, and tenderness (*p* < 0.05). Furthermore, koji-treated samples showed significant differences in CIE L* and CIE a*, which were assumed to enhance the Maillard reaction caused by an increase in the proteolytic derivative. Overall, this study provides insights into applying koji as a potential curing treatment to improve the quality of chicken breasts. Koji can be used as a novel technology in the food industry to simultaneously improve taste and tenderness. In the future, subsequent studies to investigate the effect of koji when applied in other products are needed to improve eating quality.

## Figures and Tables

**Figure 1 foods-12-03463-f001:**
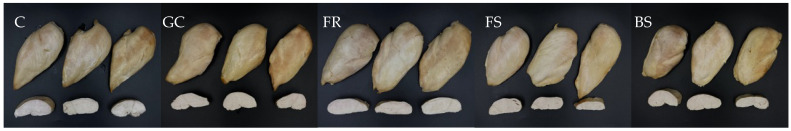
External and internal appearance of chicken breasts with different curing treatments. C, treated by superheated steam and hot air oven; GC, treated with Goldchickide as curing agent by superheated steam and hot air oven; FR, treated with curing agent as rice and fungi (*A. oryzae*) by superheated steam and hot air oven; FS, treated with curing agent as soybean and fungi (*A. oryzae*) by superheated steam and hot air oven; BS, treated with curing agent as soybean and bacteria (*B. subtilis*) by superheated steam and hot air oven.

**Table 1 foods-12-03463-t001:** Color measurement of chicken breast with different curing treatments.

Sample	CIE L*			CIE a*			CIE b*		
C	76.2	±	3.0 ^a^	4.2	±	0.9 ^b^	17.8	±	3.3
GC	74.0	±	3.7 ^ab^	4.0	±	1.7 ^ab^	20.0	±	1.8
FR	73.6	±	4.2 ^ab^	4.0	±	1.3 ^ab^	19.6	±	4.9
FS	74.8	±	2.8 ^a^	3.3	±	1.1 ^ab^	19.9	±	5.0
BS	71.7	±	3.3 ^b^	5.1	±	2.2 ^a^	19.6	±	2.9

C, treated by superheated steam and hot air oven; GC, treated with Goldchickide as curing agent by superheated steam and hot air oven; FR, treated with rice and fungi (*A. oryzae*) as curing agent by superheated steam and hot air oven; FS, treated with soybean and fungi (*A. oryzae*) as curing agent by superheated steam and hot air oven; BS, treated with soybean and bacteria (*B. subtilis*) as curing agent by superheated steam and hot air oven. CIE L*, lightness; CIE a*, redness; CIE b*, yellowness. ^a,b^ Means with different letters in a column are significantly different based on one-way ANOVA followed by Duncan’s multiple range test (*p* < 0.05).

**Table 2 foods-12-03463-t002:** Water content, cooking loss, and pH of chicken breast with different curing treatments.

Sample	Water Content			Cooking Loss			pH		
C	68.8	±	1.6	24.0	±	0.2	6.2	±	0.1 ^a^
GC	70.1	±	0.3	21.7	±	1.8	6.2	±	0.0 ^a^
FR	68.4	±	0.6	23.8	±	0.9	6.1	±	0.1 ^b^
FS	68.8	±	1.2	24.0	±	0.1	6.1	±	0.0 ^b^
BS	69.4	±	0.1	23.3	±	0.8	6.1	±	0.0 ^ab^

C, treated by superheated steam and hot air oven; GC, treated with Goldchickide as curing agent by superheated steam and hot air oven; FR, treated with rice and fungi (*A. oryzae*) as curing agent by superheated steam and hot air oven; FS, treated with soybean and fungi (*A. oryzae*) as curing agent by superheated steam and hot air oven; BS, treated with soybean and bacteria (*B. subtilis*) as curing agent by superheated steam and hot air oven. ^a,b^ Means with different letters in a column are significantly different based on one-way ANOVA followed by Duncan’s multiple range test (*p* < 0.05).

**Table 3 foods-12-03463-t003:** Texture analysis of chicken breast with different curing treatments.

Sample	Hardness (N)			Adhesiveness (mJ)			Cohesiveness			Springiness (mm)			Gumminess (N)			Chewiness (mJ)		
C	66.3	±	4.1 ^a^	1.2	±	0.7 ^b^	0.3	±	0.0 ^b^	8.0	±	0.7 ^b^	19.1	±	1.9 ^bc^	17.5	±	4.0 ^ab^
GC	61.7	±	9.6 ^ab^	2.5	±	1.0 ^ab^	0.4	±	0.1 ^a^	8.9	±	0.1 ^a^	23.7	±	2.0 ^a^	20.7	±	1.8 ^a^
FR	56.8	±	3.5 ^bc^	1.2	±	0.6 ^b^	0.4	±	0.0 ^a^	8.5	±	0.4 ^ab^	20.7	±	2.6 ^bc^	17.3	±	3.1 ^ab^
FS	52.8	±	6.5 ^c^	1.8	±	0.7 ^ab^	0.4	±	0.1 ^a^	8.3	±	0.4 ^b^	18.6	±	1.5 ^c^	16.2	±	2.2 ^b^
BS	57.7	±	8.8 ^bc^	1.8	±	0.2 ^a^	0.4	±	0.0 ^a^	8.6	±	0.1 ^ab^	21.4	±	3.1 ^ab^	20.6	±	4.5 ^a^

C, treated by superheated steam and hot air oven; GC, treated with Goldchickide as curing agent by superheated steam and hot air oven; FR, treated with rice and fungi (*A. oryzae*) as curing agent by superheated steam and hot air oven; FS, treated with soybean and fungi (*A. oryzae*) as curing agent by superheated steam and hot air oven; BS, treated with soybean and bacteria (*B. subtilis*) as curing agent by superheated steam and hot air oven. ^a–c^ Means with different letters in a column are significantly different based on one-way ANOVA followed by Duncan’s multiple range test (*p* < 0.05).

**Table 4 foods-12-03463-t004:** Intensity of chicken breasts with different curing treatments.

Sample	Appearance			Scent			Umami			Juiciness			Tenderness		
C	5.6	±	2.3	5.5	±	2.2	3.6	±	1.0 ^c^	3.2	±	1.2 ^d^	3.2	±	1.0 ^d^
GC	4.8	±	2.3	4.5	±	1.9	4.7	±	1.1 ^b^	4.3	±	1.6 ^cd^	4.0	±	1.0 ^cd^
FR	4.6	±	2.4	5.4	±	2.1	5.8	±	1.3 ^a^	5.7	±	1.3 ^ab^	5.9	±	1.5 ^ab^
FS	4.4	±	1.9	4.5	±	1.8	5.4	±	0.5 ^ab^	5.2	±	1.6 ^bc^	4.9	±	1.3 ^bc^
BS	4.6	±	2.3	5.6	±	2.6	6.3	±	0.9 ^a^	6.6	±	1.0 ^a^	6.3	±	0.9 ^a^

C, treated by superheated steam and hot air oven; GC, treated with Goldchickide as curing agent by superheated steam and hot air oven; FR, treated with rice and fungi (*A. oryzae*) as curing agent by superheated steam and hot air oven; FS, treated with soybean and fungi (*A. oryzae*) as curing agent by superheated steam and hot air oven; BS, treated with soybean and bacteria (*B. subtilis*) as curing agent by superheated steam and hot air oven. Intensities for chicken breasts with different curing treatments using a 9-point scale (“1 = extremely weak” to “9 = extremely strong”). ^a–d^ Means with different letters in a column are significantly different based on one-way ANOVA followed by Duncan’s multiple range test (*p* < 0.05).

**Table 5 foods-12-03463-t005:** Preference of chicken breasts with different curing treatments.

Sample	Appearance			Scent			Umami			Juiciness			Tenderness			Overall Preference		
C	4.3	±	1.7	5.1	±	2.6	4.0	±	1.0 ^c^	3.2	±	1.6 ^c^	2.7	±	1.2 ^c^	2.9	±	1.3 ^d^
GC	5.0	±	1.5	5.0	±	1.6	5.1	±	1.0 ^bc^	4.5	±	1.6 ^bc^	4.6	±	1.7 ^b^	3.9	±	0.9 ^c^
FR	5.0	±	1.3	5.6	±	1.2	6.0	±	1.3 ^ab^	5.9	±	1.8 ^ab^	6.0	±	0.7 ^a^	5.7	±	0.9 ^b^
FS	5.5	±	2.0	5.8	±	1.6	5.5	±	2.0 ^ab^	5.5	±	2.2 ^ab^	5.8	±	1.0 ^a^	6.7	±	0.5 ^a^
BS	5.1	±	2.3	5.9	±	1.1	6.5	±	0.9 ^a^	6.7	±	1.5 ^a^	6.6	±	1.0 ^a^	7.4	±	1.1 ^a^

C, treated by superheated steam and hot air oven; GC, treated with Goldchickide as curing agent by superheated steam and hot air oven; FR, treated with rice and fungi (*A. oryzae*) as curing agent by superheated steam and hot air oven; FS, treated with soybean and fungi (*A. oryzae*) as curing agent by superheated steam and hot air oven; BS, treated with soybean and bacteria (*B. subtilis*) as curing agent by superheated steam and hot air oven. Preference for chicken breasts with different curing treatments using a 9-point scale (“1 = extremely dislike” to “9 = extremely like”). ^a–d^ Means with different letters in a column are significantly different based on one-way ANOVA followed by Duncan’s multiple range test (*p* < 0.05).

## Data Availability

The data presented in this study are available on request from the corresponding author.

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
