# Peer review of "The Effect of Fermented Grains (koji) on Physicochemical and Sensory Characteristics of Chicken Breasts"

_foods, 2023, doi:10.3390/foods12183463_

Round 1

Reviewer 1 Report

This work presents an interesting study that explores an alternative way of curing chicken. The study could be for Foods however a few issues need to be addressed as seen below.  

In the  abstract I would suggest to change “Quality” into  “eating quality “

Where are the results from the principal component analysis mentioned in the abstract? These are mentioned but there is no graph or table showing these in the text or as supplementary material at least.

In line 50-51: “Adding inhibitors such as ascorbic acid and cysteine protease inhibitor, with these proteases to reduce the enzyme activity is an inconvenience” Can the authors describe why?

Line69: “Few studies have examined the physicochemical properties of chicken breast treated with koji”. Can the authors summarise the findings of these studies?

Goldchickide (GC) can the authors describe the characterestics of this product? Could this ingredient have higher content of salt? Could the content of salt result in differences in uptake of curing and differences in juiciness or

Overall the legends need to be more descriptive throughout the document.

How were the students selected and how were they trained for sensory evaluation?

For sensory evaluation samples were heated for 30 s, why was this duration chosen? How did the authors ensure samples were safe to be consumed?

Describe in detail how the different texture attributes were measured.

How reliable are the results of preference after the panellists had scored the other attributes? This could have an impact on the preference of the product. Moreover, where did 1 correspond and were did 9 correspond for each attribute?

How reliable is it to measure acceptance by 10-12 semi-trained panelists? The authors need to highlight this limitation.

Can the authors describe why they used the Duncan’s pairwise comparison test? To highlight that they compared all treatments with control.

Line 160: No significant differences were observed among the chicken breasts for  external and internal appearances. How was the appearance measured?

Line 201: which Koji samples had pH =5.4? all samples had pH between 6.1-6.2.

Could the authors discuss their own findings and why their findings agree with the literature findings or not, especially in the case of pH and cooking loss? While the expressible moisture and colour are discussed sufficiently.

Are there any correlations between the instrumental analysis and the sensory evaluation of the samples? It is mentioned in line 296 that there is correlation but it is not mentioned in the methods how the correlation was tested (which test was used) and what the findings were.

Overall could the authors be more descriptive in the legends? For example see below:

Table 4. The intensity of sensory attributes of chicken breasts with different curing treatments based on sensory evaluation using 9-point scales.

Table 5: Preference for chicken breasts with different curing treatments based on liking tests using 9-point scales

In the notes of the graphs could the authors write : a-d Means with different letters in a column are significantly different (p<0.05) based on … (mention the test).

In the conclusions the authors need to highlight some of the limitations of their study in relation to the sensory evaluation and they need to bring together the sensory evaluation results with the  instrumental analysis. Moreover they should suggest how the industry could take these findings further.

the quality of English is satisfactory

Reviewer 2 Report

The present work by Jeon et al explored the application of koji-fermented rice and soy grains to the quality characteristics of the chicken breast during curing. The application of koji fermented grains as a novel curing agent is the novelty of this work. The study has some significant findings and recommendations on the application of koji in curing of meat. The language is easy to understand. The hypothesis is well explained and sound.

I have the following observations-

       i.          Title: The author may mention koji in the title also, maybe fermented grains (koji) for better clarity

     ii.           Keywords: Sensory also comprises umami, tenderness; so, authors need to delete the duplications in keywords

   iii.          L32: comparatively low protein cost to red meat? Please add if correct

   iv.          L74: please mention the composition of GC if not violate copyrights

     v.           L139: tenderness and overall preference; also mention very brief about how these panelist were trained

   vi.          Statistical analysis- appropriate

  vii.          Table 1: color values are for external surface?

viii.          L236: A controversial statement; Not really related to the present work. I would suggest deleting it

Thank you for the opportunity to read your work.

Round 2

Reviewer 1 Report

Dear authors, 

I am happy with the changes made to the manuscript. 

My only concern is the sentence added to all the legends now .

"Experiments were performed in triplicate and repeated three times with similar results." What do they authors mean by similar results? How did they test the results were similar? Does it mean they had 3*3 replicates in other words n=9 replicates or n=3 replicates? 

There are some sentences that need to be checked like the last sentence added to the conclusions section
